# More harmful, less noticeable: Learning Adversarial Null-Text Embeddings for Inconspicuous Attack

## Abstract

Adversarial examples, which are artificially crafted data intended to disrupt the output of deep learning models, present a new round of challenges to the stability and security of artificial intelligence technology. Unrestricted adversarial examples, obtained by modifying the semantic elements of images, have the characteristics of being natural and semantically meaningful. However, previous methods either significantly altered the image's color or content, or blurred visual details (such as text or geometric designs), making the generated adversarial examples easily detectable by the human eye. In this paper, we propose a method to generate highly natural adversarial examples based on stable diffusion. This is achieved by introducing adversarial loss during the image reconstruction process to perturb cross-attention mechanism. To further enhance image quality, we introduce perceptual loss into the adversarial attack process for the first time. Extensive experiments and visualizations demonstrate the effectiveness of our proposed method. Compared to the current state-of-the-art methods, our approach not only improves the adversarial transferability by an average of $12.59 - 50.3\%$ but also significantly enhances image quality. Code will be publicly available.

## 1 Introduction

The development of deep learning technologies has driven advancements across various industries. However, past research has revealed inherent limitations in deep learning models, particularly neural networks. Specifically, their performance significantly degrades when confronted with carefully crafted adversarial samples, a phenomenon known as adversarial attacks, and the resulting special samples are referred to as adversarial examples (Szegedy et al., 2014; Goodfellow et al., 2014; Moosavi-Dezfooli et al., 2016). Research on adversarial attacks and defenses offers a novel perspective for studying neural networks, contributing to their interpretability and robustness (Ilyas et al., 2019; Zhang et al., 2020). Early adversarial attack methods manipulated image pixel values based on gradient information to perturb the target model's output (Madry et al., 2018; Su et al., 2019). Since the magnitude of pixel perturbations was constrained by the $L_p$-norm, these methods produced what are referred to as restricted adversarial examples. Although these methods enable effective white-box attacks, their noise-like fixed patterns have made it feasible to develop targeted defense strategies, including data preprocessing Liao et al. (2018); Xie et al. (2018); Nie et al. (2022) and adversarial training (Tramèr et al., 2018; Shafahi et al., 2019).

In contrast to the aforementioned methods, recent approaches advocate for generating "natural" adversarial examples by manipulating semantic elements of images (e.g., color, texture) to produce data that significantly reduce the confidence of deep learning models (Zhao et al., 2020; Shamsabadi et al., 2020). This is referred to as unrestricted adversarial examples, meaning they are not subject to $L_p$-norm constraints. Compared to perturbation-based examples, unrestricted adversarial examples follow a different pattern: they exhibit high transferability without requiring meticulously designed methods Yuan et al. (2022); Chen et al. (2024) and are more effective at bypassing defenses aimed at $L_p$-norm perturbations (Chen et al., 2023). This presents new challenges for researchers focused on model robustness and defense strategies targeting $L_p$-norm adversarial examples. Given that unrestricted adversarial examples offer similar (or even greater) value than restricted examples, this

paper focuses on the study of unrestricted adversarial examples, calling for increased attention from researchers in this domain.

Unrestricted adversarial examples often focus on manipulating the colors of the original image. This includes transformations of the entire image's color space Hosseini & Poovendran (2018), dividing the image into regions and applying different degrees of color changes Yuan et al. (2022), and using coloring models for more flexible color editing (Bhattad et al., 2019). Subsequent research has attempted attack methods based on texture and style transfer (Bhattad et al., 2019; Liang & Xiao, 2023). However, focusing solely on color or texture elements to some extent hinders the degree to which adversarial examples are unrestricted. This undoubtedly weakens the effectiveness and transferability of adversarial examples. With the rapid development of generative AI, some research have attempted to introduce generative models into the creation of unrestricted adversarial examples. This includes methods based on class labels Song et al. (2018); Dai et al. (2023) and content-based methods (Chen et al., 2024; 2023). Generative model-based methods are not limited to specific color or texture elements, allowing them to produce more diverse and effective adversarial examples.

Although intuitively, it seems reasonable that achieving high transferability comes at the cost of altering more content in the original image, we still tried to explore in this paper: Is it possible to get higher transferability by less cost? Specifically, we aim to achieve the following three objectives: 1) Modify a given image based on a diffusion model to produce unrestricted adversarial examples; 2) The generated results should exhibit the characteristics of unrestricted adversarial examples, which are high transferability and effectiveness against common defense methods; 3) The generated results should have minimal content alteration compared to the original image. Therefore, in this paper, we propose a framework for generating unrestricted adversarial examples based on latent diffusion model (Rombach et al., 2022). Unlike previous approaches that directly optimize latent space features, we draw inspiration from recent work on controllable generation Mokady et al. (2023); Meng et al. (2021); Kawar et al. (2023) and achieve fine-grained control over the generated results by optimizing null-text embeddings during the denoising process, thereby significantly improving the quality of the generated examples. In conclusion, our main contributions are:

- We propose a method for generating unrestricted adversarial examples based on LDMs. Our method requires only minimal modifications to the image content, effectively addressing the challenge that previous methods faced in balancing high transferability and image fidelity.

- We optimize the null-text embedding by introducing adversarial loss at specific denoising steps during the image reconstruction process, which can subtly alter the unconditional generative process and achieve inconspicuous attack.

- We incorporate perceptual loss into the generation of unrestricted adversarial examples, significantly improving visual effect and achieving state-of-the-art image quality with higher transferability.

## 2 RELATED WORKS

### 2.1 $L_p$-NORM ADVERSARIAL EXAMPLES

Since the seminal work of Christian et al. Szegedy et al. (2014), there has been extensive discourse on methods for executing white-box attacks based on the gradient information of the target model Goodfellow et al. (2014); Madry et al. (2018); Moosavi-Dezfooli et al. (2016); Tramèr et al. (2018). These strategies typically require the initial input of pristine samples into the target network to derive the predicted probability distribution. Su et al. (2019) achieved adversarial objectives by modifying a single pixel, thereby generating adversarial examples with heightened deceptive characteristics. Carlini & Wagner (2017) introduced a C&W algorithm, which proved effective in undermining defensive distillation mechanisms. To alleviate the complexity of optimization techniques, subsequent research Xiao et al. (2018); Baluja & Fischer (2018); Jandial et al. (2019) proposed the use of neural networks to model the transformation mapping from clean to adversarial samples, thereby enhancing efficiency. However, $L_p$ norm-based adversarial examples are merely numerical changes obtained through optimization strategies. They fail to reveal which semantic features of the input data pose blind spots for deep learning models, and they can be easily filtered out by defenses designed specifically for $L_p$-norm perturbation.

## 2.2 UNRESTRICTED ADVERSARIAL EXAMPLES

To break free from the constraints of $L_p$ norm, some studies have attempted to solve adversarial examples in different search spaces. Zhao et al. (2018) uses GANs to map input data to a latent space and searches for adversarial examples near the latent features. Hosseini & Poovendran (2018) converts the original RGB image to the HSV space, then globally modifies the H and S values while keeping V unchanged, claiming this preserves the basic shape of the image. Alaifari et al. (2019) proposes iteratively applying small deformations to the image plane based on a small amplitude vector field to create adversarial examples. Bhattad et al. (2019) proposes methods for attacking by recoloring or transferring textures of the image. Shamsabadi et al. (2020) excludes areas of the image that are easily perceptible to humans and applies color changes based on prior intuition. Furthermore, Yuan et al. (2022) transform clean images to adversarial variants with realistic color distribution sampled from ADE20K dataset Zhou et al. (2019). However, the aforementioned methods all rely on carefully designed modification patterns, including color, texture, human intuition, and specific data distributions. These constraints mean that the generated adversarial examples are not as "unrestricted" as their names suggest. Therefore, some studies have attempted to use generative models to achieve more flexible and diverse unrestricted adversarial attacks. Song et al. (2018) models the class-conditional distribution on data samples by training an AC-GAN, and then searches the latent space for samples that may cause the target classifier to fail under the given class. Dai et al. (2023) extends this idea by introducing the latent diffusion model Rombach et al. (2022), which generates false negative samples with better generative performance for the corresponding class. Xue et al. (2024) utilizes the SDEdit method to purify the obtained adversarial examples in a multi-step adversarial attack, resulting in outcomes that lie in the intersection of the support sets of the adversarial example distribution and the real data distribution. To improve the consistency between adversarial examples and the original images, Chen et al. (2024) maps input samples onto a low-dimensional manifold via latent image mapping and then uses adversarial latent optimization to guide Stable Diffusion to generate adversarial results. Chen et al. (2023) enhances adversarial transferability by disrupting the original cross-attention maps and maintains content similarity through self-attention control and a reduced number of DDIM inversion steps. However, methods based on Stable Diffusion typically focus on directly optimizing latent features, and the impact of optimizing conditional inputs has yet to be thoroughly studied.

## 3 METHODOLOGY

### 3.1 PROBLEM DEFINITION

Given a clean image $x_0$ with a true label $y_0$, adversarial attacks aim to find an adversarial example $x_{adv}$ that is highly similar to $x_0$ but misleads the classifier such that its predicted output $y'$ is not equal to $y_0$. Methods based on the $L_p$-norm add a small perturbation $\delta$ to the original input, then optimize the value of based on the gradient of the target classifier and restrict it within a certain range $\kappa$ to maximize the cross-entropy loss:

$$\max_{\delta} \mathcal{L}_{ce}(\mathcal{F}(x_{adv} = x_0 + \delta),\ y_0) \quad s.t. \quad \| \delta \|_p < \kappa, \tag{1}$$

On the other hand, unrestricted adversarial examples typically use image transformation methods to semantically manipulate the clean image and then adjust the transformation strategy based on the gradient of classifier, which means:

$$\max_{\theta} \mathcal{L}_{ce}(F(x_{adv} = \mathcal{G}(x_0, \theta)),\ y_0) \tag{2}$$

where $\mathcal{G}$ can be color space transformation algorithms, colorization models or generative models, and $\theta$ are parameters that can be optimized. Recent research on using diffusion models for adversarial transformations of images has demonstrated strong transferability. However, the generated images exhibit poor visual quality. Therefore, we aim to design more refined adversarial transformation strategies by controlling the generation results based on null-text embeddings instead of latent maps.

## 3.2 PERTURB LATENT FEATURE BY CROSS-ATTENTION

Latent diffusion models(LDMs) progressively denoise and generate realistic images by iteratively predicting the noise present in the latent features at each timestep $t$:

$$\tilde{\epsilon}_\theta(z_t, t, \mathcal{C}, \varnothing) = w \cdot \epsilon_\theta(z_t, t, \mathcal{C}) + (1 - w) \cdot \epsilon_\theta(z_t, t, \varnothing) \tag{3}$$

where $z_t$ is latent features, $\mathcal{C}$ and $\varnothing$ are text embedding and null-text embedding, and $w$ is a guidance factor. To make the generated results adversarial, perturbations can be added to the latent features and optimized accordingly. However, applying global perturbations to the latent features tends to obscure the original visual details, even when starting from intermediate timesteps, as demonstrated in previous studies (Chen et al., 2024; 2023). In fact, the textual embeddings input at each timestep can also influence the generated results through the cross-attention mechanism.

$$\text{Attention}(Q, K, V) = \text{softmax}(\frac{QK^T}{\sqrt{d}}) \cdot V \tag{4}$$

$$Q = W_Q^{(i)} \cdot \phi_i(z_t), K = W_K^{(i)} \cdot \psi(y), V = W_V^{(i)} \cdot \psi(y) \tag{5}$$

Here, $\phi_i(z_t) \in \mathbb{R}^{N \times d_\epsilon^i}$ represents the flattened output of the UNet, and $W_Q$, $W_K$ and $W_V$ are learnable projection matrices. $\psi(\cdot)$ is a pretrained text encoder. We influence the attention mechanism by introducing a perturbation $\delta$ to the textual embedding so that $\psi(y) \to \psi(y) + \delta$. Then the attention result can be reformulated as:

$$\text{Attention} = \text{softmax}\left( \frac{Q(W_K^{(i)} \cdot \psi(y))^T + Q(W_K^{(i)} \cdot \delta)}{\sqrt{d}} \right) \cdot \left( W_V^{(i)} \cdot \psi(y) + W_V^{(i)} \cdot \delta \right) \tag{6}$$

Due to the complexity of the latent feature space, directly adding perturbations may result in non-linear amplification effects, leading to instability in the generated outputs. In contrast, applying perturbations to the textual embedding is more controllable, as its influence is smoothly propagated to global features through the cross-attention mechanism, without introducing intense localized noise.

## 3.3 LEARNING ADVERSARIAL NULL-TEXT EMBEDDING

Perturbations to the cross-attention results can be achieved by introducing perturbations to either the text embedding or the null-text embedding. However, since the text embedding is highly correlated with the content of the generated image, we opted to perturb the null-text embedding instead. We achieve this goal by utilizing DDIM inversion and null-text optimization (Mokady et al., 2023). Given a clean image $x_0$ and corresponding text description $\mathcal{P}$, pre-trained VAE project it into a low-dimension feature map $z_0$ in latent space. Base on the assumption that the ordinary differential equation (ODE) process can be reversed in the limit of small steps, DDIM inversion process can be formulated as :

$$z_{t+1} = \sqrt{\frac{\alpha_{t+1}}{\alpha_t}} z_t + \sqrt{\alpha_{t+1}}(\sqrt{\frac{1}{\alpha_{t+1}} - 1} - \sqrt{\frac{1}{\alpha_t} - 1}) \cdot \epsilon_\theta(z_t, t, \mathcal{C}) \tag{7}$$

here $\mathcal{C} = \psi(\mathcal{P})$. Theoretically, after $T$ steps of adding noise, DDIM inversion can yield an initial noise $z_T^*$, from which one can progressively denoise and restore the original image $z_0$ based on Eq.3. However, the default value of $w$ is set to 7.5 for Stable Diffusion. This is inconsistent with the forward process based on DDIM inversion, as Eq.7 indicates that the noise is predicted based on the prompt $\mathcal{C}$. Therefore, the null-text embeddings input at each sampling step need to be optimized to make the network output close to the results obtained by DDIM inversion at the corresponding steps:

$$\min_{\varnothing_t} \| \bar{z}_{t-1} - z_{t-1}^* \|_2^2, \tag{8}$$

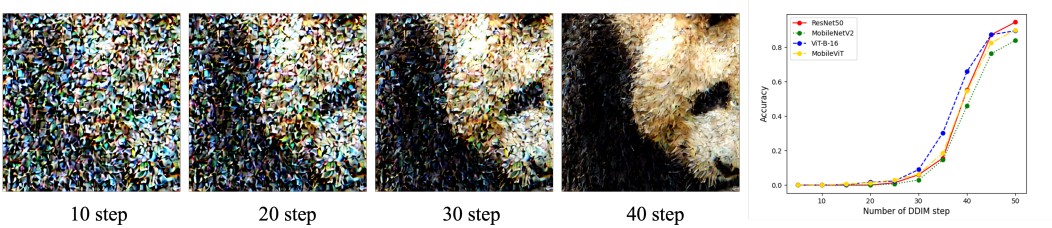

Figure 1: The intermediate results of DDIM. We can see that in the first 20 steps, there are so many noises in the generated result that can mislead classifier. And the accuracy of the target classifier is really very low. This means that adding adversarial gradient to the first several steps is meaningless.

$$\bar{z}_{t-1} = z_{t-1}(\bar{z}_t, t, \varnothing_t, \mathcal{C}) \tag{9}$$

For simplicity $z_{t-1}(\bar{z}_t, t, \varnothing_t, \mathcal{C})$ denotes applying DDIM sampling step using $\bar{z}_t$ under the unconditional embedding $\varnothing_t$ and the conditional embedding $\mathcal{C}$:

$$z_{t-1}(\bar{z}_t, t, \mathcal{C}, \varnothing_t) = \sqrt{\frac{\alpha_{t-1}}{\alpha_t}}\bar{z}_t + \sqrt{\alpha_{t-1}}\left(\sqrt{\frac{1}{\alpha_{t-1}} - 1} - \sqrt{\frac{1}{\alpha_t} - 1}\right) \cdot \tilde{\epsilon}_\theta(\bar{z}_t, t, \mathcal{C}, \varnothing_t) \tag{10}$$

During the null-text optimization process, we directly introduce adversarial perturbations to the null-text embeddings. The reason for choosing null-text embeddings over text embeddings as the optimization target is that text embeddings significantly affect the content of the generated image, whereas the impact of null-text embeddings is less perceptible. Specifically, we incorporate the classifier to be attacked into the null-text optimization process at the sampling step $t$. The optimization objective is expressed as follows:

$$\min_{\varnothing_t} \beta \cdot \mathcal{L}_{mse}(\bar{z}_{t-1}, \; z_{t-1}^*) - \mathcal{L}_{ce}(\mathcal{F}(\mathcal{D}(\bar{z}_{t-1})), \; y) \tag{11}$$

Here, $\mathcal{D}$ is the decoder of the VAE, and $y$ is the true label of the image $x_0$. However, null-text optimization process usually takes 500 steps, which makes it very time-consuming. We delay the injection of the adversarial gradient to further reduce the extent of modification to the original image. This approach not only saves runtime but also does not affect the adversarial nature of the generated results, because, as shown in Figure.1, when $t$ approaches $T$, the generated results still contain excessive noise, causing the classifier to naturally misclassify them. We measured the classifier's accuracy at different time steps and decided to add the misclassification loss at the last 10 DDIM steps.

### 3.4 IMPROVING PERCEPTUAL QUALITY

Through the aforementioned process, we obtain a set of adversarial null-text embeddings $\{\varnothing_t^{adv}\}_{t=1}^T$ that can control the Stable Diffusion model to generate examples that are faithful to the original image while also being adversarial. However, we observed that the results obtained by this method lack high-frequency features, specifically manifesting as a certain degree of local blurriness. We believe this is due to the limitations of the $L_2$ loss constraint. It is well-known that $L_2$ loss in the latent space is not a good measure of image fidelity; it can accurately capture low frequencies but fails to promote high-frequency clarity. Hence, the results generated by VAE tend to be somewhat blurry. Diffusion models mitigate the blurring effect caused by $L_2$ loss by increasing the number of sampling steps $T$, thereby reducing the difference between intermediate results at each step. However, our method significantly modifies the output results at each step during the latter stages of DDIM sampling, which exacerbates the negative impact of $L_2$ loss. To address this issue, we introduce perceptual loss Zhang et al. (2018) into the optimization objective to enhance the restoration of high-frequency details. Feature reconstruction perceptual loss calculates the similarity between the input image $\hat{x}$ and the target image $x$ in the feature space using a network $\mathcal{G}$ to map both images into this space:

$$\mathcal{L}_{feat}^{\mathcal{G},j}(\hat{x}, x) = \frac{1}{C_j H_j W_j} \parallel \mathcal{G}_j(\hat{x}) - \mathcal{G}_j(x) \parallel_2^2 \tag{12}$$

---

**Algorithm 1** Example Algorithm

---

1: **Input:** an input image $z_0$ with label $y$, a corresponding text embedding $\mathcal{C} = \psi(\mathcal{P})$, a classifier $\mathcal{F}_\theta(\cdot)$, DDIM steps $T$, null-text optimization iteration $N$, and adversarial start timestep $t_s^{adv}$.
2: Calculate latents $\{z_0^*, ..., z_T^*\}$ using Equation.7 over $z_0$
3: Initialize variables
4: **for** $t = t_s^{adv}, t_s^{adv} - 1, ..., 1$ **do**
5:    **for** $j = 0, ..., N - 1$ **do**
6:       $\varnothing_t \leftarrow \varnothing_t - \eta \nabla_\varnothing \mathcal{L}(z_{t-1}(\bar{z}_t, \varnothing_t, \mathcal{C}), z_{t-1}^*);$
7:    **end for**
8:    Set $\bar{z}_{t-1} \leftarrow z_{t-1}(\bar{z}_t, \varnothing_t, \mathcal{C}), \varnothing_{t-1} \leftarrow \varnothing_t;$
9: **end for**
10: **Output:** The unrestricted adversarial example $\bar{x}_0 = \mathcal{D}(\bar{z}_0)$.

---

Here $\mathcal{G}_j$ represents the activation of the $j$-th layer in network $\mathcal{G}$. If $j$ is a convolutional layer then $\mathcal{G}_j(x)$ will be a feature map of shape $C_j \times H_j \times W_j$. After introducing feature reconstruction perceptual loss, the final optimization object is as follows:

$$\mathcal{L}(\bar{z}_{t-1}, z_{t-1}^*) = \beta \cdot \mathcal{L}_{mse}(\bar{z}_{t-1},\ z_{t-1}^*) + \gamma \cdot \mathcal{L}_{feat}^{\mathcal{G}}(\mathcal{D}(\bar{z}_{t-1}), x_0) - \mathcal{L}_{ce}(\mathcal{F}(\mathcal{D}(\bar{z}_{t-1})), y) \quad (13)$$

$\gamma$ is a scaling factor used to adjust the weight of the perceptual loss. In this paper, we set $\gamma = 0.1$. The perceptual loss is calculated based on the VGG16 model pre-trained on the ImageNet dataset. We use the outputs of the 4-th, 9-th, 16-th, and 23-rd layers of the VGG16 network to compute feature similarity.

## 4 EXPERIMENTS

### 4.1 EXPERIMENTAL SETUP

**Dataset.** Our experiments are conducted on the ImageNet-compatible Dataset (Kurakin et al., 2018b). The dataset consists of 1,000 images from the validation set of ImageNet, and is widely used in recent adversarial attacking research (Xie et al., 2019; Gao et al., 2020; Dong et al., 2019; Yuan et al., 2022; Chen et al., 2024).

**Attack Evaluation.** We selected SAE Hosseini & Poovendran (2018), ColorFool Shamsabadi et al. (2020), ACE Zhao et al. (2020), NCF Yuan et al. (2022), ACA Chen et al. (2024) and DiffAttack Chen et al. (2023) as our comparison methods. The evaluation criterion for adversarial performance is the attack success rate (ASR), which is the proportion of samples misclassified by the target model out of all samples. The average attack success rate (Ave.ASR) is consistent with Chen et al. (2024), which is the average attack success rate on non-surrogate models.

**Models.** To fully measure the performance of our method compared to other methods across various classification models, we selected models with different architectures, including convolutional neural networks (CNNs) and Transformers, as attack targets. For CNNs, we chose ResNet-50 (RN-50) He et al. (2016), ResNet152 (RN-152) He et al. (2016), MobileNet-v2 (MN-v2) Sandler et al. (2018), DenseNet-161 (Dense-161) Huang et al. (2017), EfficientNet-b7 (EF-b7) Tan & Le (2019), and Inception-v3 (Inc-v3) Szegedy et al. (2016). For Transformers, we selected ViT-Base-16 (ViT-B) Dosovitskiy et al. (2020), MobileViT-small (MobViT-s) Sandler et al. (2018), Swin-Transformer-Base (Swin-B) Liu et al. (2022), and Pyramid Vision Transformer (PVT-v2) Wang et al. (2022). For a fair comparison, all models are evaluated using their respective default input sizes and normalization coefficients in PyTorch.

**Implementation Details.** Our experiments are conducted on a single NVIDIA 6000 Ada GPU. DDIM steps= 50, adversarial injection starting timestep $t_s^{adv} = 10$, $\beta = 30.0$, $\gamma = 0.1$. We selected Adam Kingma & Ba (2014) as optimizer. Learning rate is $1e-2$. For a fair comparison, all methods based on Stable Diffusion use the SDv1.4 checkpoint. Corresponding prompts are generated based on BLIP v2 Li et al. (2023) automatically.

Table 1: Performance comparison of adversarial transferability on normally trained CNNs and ViTs. We report attack success rates (%) of each method. ("*" means white-box attack results. Red text and underline text represent the best and second best result, respectively.)

| Surrogate Model | Attack | Models | | | | | | | | | | Avg. ASR(%) |
| | | CNNs | | | | | | Transformers | | | | |
| | | RN-50 | Inc-v3 | MN-v2 | Dense-161 | RN-152 | EF-b7 | MobViT-s | ViT-B | Swin-B | PVT-v2 | |
| --- | --- | --- | --- | --- | --- | --- | --- | --- | --- | --- | --- | --- |
| - | Clean | 5.60 | 7.60 | 13.60 | 7.10 | 3.20 | 4.90 | 7.10 | 7.70 | 4.60 | 4.10 | 6.55 |
| RN-50 | SAE | 71.70* | 21.60 | 42.70 | 36.20 | 24.90 | 19.20 | 35.10 | 32.10 | 25.00 | 16.90 | 28.19 |
| | ColorFool | 71.40* | 19.20 | 41.70 | 33.70 | 16.10 | 12.60 | 27.80 | 22.40 | 13.20 | 10.20 | 21.88 |
| | ACE | 91.80* | 18.90 | 34.20 | 21.30 | 10.30 | 12.00 | 24.00 | 13.80 | 9.70 | 5.70 | 16.66 |
| | NCF | 78.10* | 46.30 | 68.80 | 53.30 | 40.20 | 37.80 | 55.60 | 43.20 | 32.40 | 21.90 | 44.39 |
| | ACA | 72.70* | 56.10 | 60.30 | 56.90 | 53.30 | 53.50 | 57.60 | 51.50 | 52.70 | 47.50 | 54.38 |
| | DiffAttack | 79.30* | 46.10 | 49.00 | 49.20 | 46.10 | 45.90 | 48.10 | 38.70 | 40.80 | 37.70 | 44.62 |
| | **Ours** | 94.40* | 70.30 | 74.10 | 74.70 | 70.80 | 63.80 | 70.80 | 57.60 | 65.40 | 55.20 | 66.97 |
| MN-v2 | SAE | 30.50 | 25.00 | 86.30* | 40.10 | 27.60 | 22.90 | 37.00 | 34.00 | 25.40 | 16.60 | 28.79 |
| | ColorFool | 15.40 | 13.80 | 84.20* | 23.70 | 10.20 | 8.70 | 18.00 | 14.90 | 8.40 | 7.20 | 13.37 |
| | ACE | 10.00 | 17.40 | 96.40* | 16.10 | 6.90 | 10.30 | 19.20 | 10.60 | 7.70 | 5.70 | 11.54 |
| | NCF | 42.20 | 47.90 | 91.10* | 53.60 | 34.30 | 39.10 | 56.00 | 42.20 | 28.30 | 19.40 | 40.33 |
| | ACA | 50.70 | 54.90 | 84.90* | 53.80 | 45.70 | 50.20 | 56.20 | 50.70 | 47.50 | 44.40 | 50.46 |
| | DiffAttack | 41.20 | 47.60 | 91.80* | 48.50 | 33.10 | 42.70 | 51.80 | 35.70 | 38.00 | 32.50 | 41.23 |
| | **Ours** | 66.50 | 66.80 | 98.80* | 70.30 | 52.70 | 54.40 | 74.80 | 54.10 | 55.30 | 45.30 | 60.02 |
| ViT-B | SAE | 27.80 | 22.50 | 41.30 | 38.00 | 22.60 | 19.80 | 37.20 | 72.40* | 24.90 | 16.50 | 27.84 |
| | ColorFool | 21.80 | 20.60 | 42.20 | 35.00 | 15.10 | 12.20 | 28.00 | 76.70* | 14.20 | 11.00 | 22.23 |
| | ACE | 13.10 | 22.40 | 33.40 | 22.90 | 8.90 | 13.10 | 25.80 | 97.60* | 10.40 | 7.80 | 17.53 |
| | NCF | 44.60 | 46.20 | 64.60 | 52.80 | 35.90 | 41.00 | 55.60 | 77.50* | 35.60 | 24.60 | 44.54 |
| | ACA | 62.50 | 66.90 | 68.90 | 66.20 | 58.20 | 58.40 | 66.30 | 87.40* | 62.10 | 55.00 | 62.72 |
| | DiffAttack | 59.30 | 60.10 | 62.50 | 59.80 | 55.50 | 58.80 | 67.60 | 84.70* | 68.00 | 62.20 | 61.53 |
| | **Ours** | 69.50 | 72.10 | 75.20 | 77.10 | 64.70 | 61.10 | 74.80 | 94.90* | 74.80 | 63.40 | 70.30 |
| Inc-v3 | SAE | 23.60 | 64.80* | 38.90 | 33.30 | 18.30 | 15.60 | 33.70 | 28.30 | 19.50 | 13.80 | 25.00 |
| | ColorFool | 9.60 | 93.70* | 21.40 | 14.30 | 5.40 | 6.10 | 14.10 | 11.20 | 5.60 | 4.60 | 10.26 |
| | ACE | 9.40 | 93.50* | 26.10 | 16.00 | 6.70 | 7.80 | 14.50 | 11.50 | 6.10 | 6.00 | 11.57 |
| | NCF | 34.90 | 75.50* | 55.40 | 41.70 | 25.70 | 23.50 | 45.50 | 34.50 | 23.10 | 16.70 | 33.44 |
| | ACA | 54.00 | 90.50* | 63.00 | 61.90 | 53.50 | 52.60 | 59.70 | 56.50 | 53.10 | 49.90 | 56.02 |
| | DiffAttack | 32.10 | 70.20* | 41.80 | 41.20 | 25.00 | 31.20 | 35.70 | 30.30 | 27.00 | 24.20 | 32.06 |
| | **Ours** | 56.10 | 97.90* | 63.80 | 67.30 | 45.30 | 50.10 | 59.30 | 45.60 | 47.40 | 37.90 | 52.53 |
| Swin-B | SAE | 31.10 | 22.30 | 44.40 | 39.30 | 24.00 | 20.20 | 37.70 | 37.80 | 69.80* | 19.10 | 30.66 |
| | ColorFool | 16.70 | 17.20 | 35.80 | 26.70 | 12.20 | 10.30 | 24.30 | 20.30 | 73.30* | 8.10 | 19.07 |
| | ACE | 16.10 | 17.40 | 31.90 | 23.70 | 11.00 | 10.20 | 21.50 | 17.60 | 96.20* | 9.50 | 17.64 |
| | NCF | 40.10 | 31.30 | 53.40 | 42.10 | 33.90 | 27.50 | 44.80 | 43.40 | 67.20* | 26.70 | 38.13 |
| | ACA | 62.50 | 62.00 | 68.00 | 64.90 | 58.80 | 59.20 | 63.20 | 62.70 | 77.90* | 59.80 | 62.34 |
| | DiffAttack | 55.60 | 53.80 | 58.50 | 52.90 | 51.20 | 60.40 | 60.70 | 57.10 | 82.60* | 65.40 | 59.82 |
| | **Ours** | 74.80 | 72.40 | 73.50 | 74.00 | 70.10 | 72.20 | 77.20 | 73.00 | 91.50* | 73.10 | 75.18 |

## 4.2 ADVERSARIAL PERFORMANCE ANALYSIS

Table 1 shows the performance comparison between our method and baseline methods on different classifiers. We selected RN50, MNV2, ViT-Base-16, Swin-Transformer-Base and Inception-v3 as surrogate models in the white-box attack scenario, and then transferred the generated adversarial examples to other classifiers that did not access gradient information to perform black-box attacks.

We first focus on the results of white-box attacks. It can be observed that unrestricted adversarial examples typically achieve only around a 70% attack success rate in white-box attack scenarios. This is because unrestricted adversarial attacks are limited by factors such as color range or segmented regions, meaning they may not always find a local optimum. However, our proposed method significantly addresses this issue, delivering a stable performance above 90% ASR across various surrogate models in white-box attack scenarios. This demonstrates that our method enables the model to learn adversarial null-text embeddings and generate highly adversarial examples.

Even more exciting are the black-box attack results. When transferring adversarial examples generated based on the surrogate models to other classifiers, the results generated by our model maintained a high attack success rate, comparable to the current state-of-the-art methods. Specifically, when using RN50 and MN-v2 as surrogate models, our method outperformed other baselines by $12.59\% - 50.31\%$ and $9.56\% - 48.48\%$, respectively. While with ViT-B and Swin-B as surrogate models, it also surpassed other baselines by $7.58\% - 52.77\%$ and $12.84\% - 57.54\%$. This demonstrates that our proposed method is capable of generating highly transferable adversarial examples when faced with both CNN and Transformer-based surrogate models.

Table 2: Image quality comparison of adversarial examples on different surrogate models. Here "FR" and "NR" refer to full-reference and no-reference image quality assessment metrics, respectively. ( Red text and underline text represent the best and second best result, respectively.)

| | RN50 | | | MN-v2 | | | ViT-B | | | Inc-v3 | | | Swin-B | | |
|---|---|---|---|---|---|---|---|---|---|---|---|---|---|---|---|
| | PSNR↑ | SSIM↑ | LPIPS↓ | PSNR↑ | SSIM↑ | LPIPS↓ | PSNR↑ | SSIM↑ | LPIPS↓ | PSNR↑ | SSIM↑ | LPIPS↓ | PSNR↑ | SSIM↑ | LPIPS↓ |
| SAE | 15.731 | 0.657 | 0.365 | 14.951 | 0.593 | 0.368 | 14.795 | 0.631 | 0.385 | 22.985 | 0.742 | 0.339 | 18.650 | 0.697 | 0.386 |
| ColorFool | 14.788 | 0.665 | 0.348 | 17.912 | 0.729 | 0.280 | 14.250 | 0.646 | 0.363 | 21.034 | 0.681 | 0.273 | 15.981 | 0.674 | 0.333 |
| NCF | 15.420 | 0.651 | 0.437 | 15.096 | 0.640 | 0.450 | 15.082 | 0.639 | 0.443 | 14.965 | 0.637 | 0.444 | 15.430 | 0.661 | 0.430 |
| ACA | 16.080 | 0.497 | 0.422 | 16.243 | 0.503 | 0.421 | 14.013 | 0.397 | 0.486 | 17.689 | 0.569 | 0.425 | 17.957 | 0.580 | 0.416 |
| DiffAttack | 22.427 | 0.640 | 0.194 | 22.381 | 0.639 | 0.197 | 21.736 | 0.621 | 0.218 | 22.379 | 0.640 | 0.194 | 22.198 | 0.633 | 0.203 |
| **Ours** | 23.935 | 0.759 | 0.183 | 23.592 | 0.748 | 0.198 | 23.059 | 0.734 | 0.210 | 23.625 | 0.748 | 0.189 | 23.649 | 0.747 | 0.191 |

Table 3: Performance comparison of adversarial methods on defense algorithm. ( Red text and underline text represent the best and second best result, respectively.)

| Attack | HGD | R&P | DiffPure | Shape-Res50 | Adv-Inc-V3 | Inc-V3$_{ens3}$ | Inc-V3$_{ens4}$ | IncRes-V2$_{ens}$ |
|---|---|---|---|---|---|---|---|---|
| SAE | 38.20 | 40.90 | 38.80 | 41.20 | 17.00 | 16.70 | 18.90 | 12.20 |
| ColorFool | 34.80 | 34.20 | 41.60 | 40.40 | 13.90 | 18.10 | 21.60 | 12.70 |
| NCF | 62.50 | 64.10 | 50.10 | 53.90 | 31.70 | 33.80 | 36.90 | 27.90 |
| ACA | 61.10 | 63.70 | 53.60 | 55.50 | 55.20 | 56.00 | 55.30 | 49.10 |
| DiffAttack | 72.80 | 69.20 | 46.30 | 48.10 | 40.20 | 38.60 | 42.10 | 33.10 |
| **Ours** | 87.90 | 86.90 | 69.90 | 68.70 | 62.00 | 58.80 | 61.10 | 52.40 |

Notably, when using Inc-v3 as the surrogate model, the average attack success rate of our method is slightly lower than that of ACA Chen et al. (2024), especially when transferring to Transformer-based models. This suggests that adversarial examples generated by our method targeting Inception-V3 are prone to local dependency overfitting. In contrast, ACA globally alters the image content, which makes it more effective when transferred to Transformer-based models.

## 4.3 IMAGE QUALITY ASSESSMENT

When converting clean samples into adversarial examples, we must not only focus on the attack performance but also on the quality of the generated images. Here, we use PSNR, SSIM Wang et al. (2004), and LPIPS Zhang et al. (2018) to measure the similarity between the generated results and the original images. Results are represented in Table 2. Since our method only modifies the high-frequency details of the image, the quality of the generated results shows significant improvement compared to previous methods. It can be observed that the PSNR of adversarial examples generated by our method remains above 23 dB in all cases, while LPIPS stays below 0.2 in most cases.

Figure 2 presents the visual comparison of our method with several baseline methods. It can be seen that unrestricted adversarial attack methods, such as NCF Yuan et al. (2022), primarily rely on altering the color of the original image, leading to unnatural color distortions that are easily noticeable by the human eye. ACA Chen et al. (2024) generated some results that are semantically consistent with the original images, but sometimes the differences are significant enough to be easily identified when a reference image (the original image) is available. Although the results generated by DiffAttack Chen et al. (2023) resemble the original image in general, a closer inspection reveals that finer geometric features tend to become distorted, such as the text on the cardboard box in the first row, which becomes difficult to read. In contrast, the results generated by our method preserve the visual details of the original image, making them less noticeable to the human eye.

## 4.4 ADVERSARIAL PERFORMANCE ON DEFENSE METHODS

As previously mentioned, current defense methods against adversarial attacks primarily focus on $L_p$-norm attacks, including input preprocessing and adversarial training. However, unrestricted adversarial examples differ significantly from $L_p$-norm examples, making them harder to defend against using existing methods. Here, we select several defense strategies that have been proven effective against $L_p$-norm adversarial examples Liao et al. (2018); Xie et al. (2018); Tramèr et al.

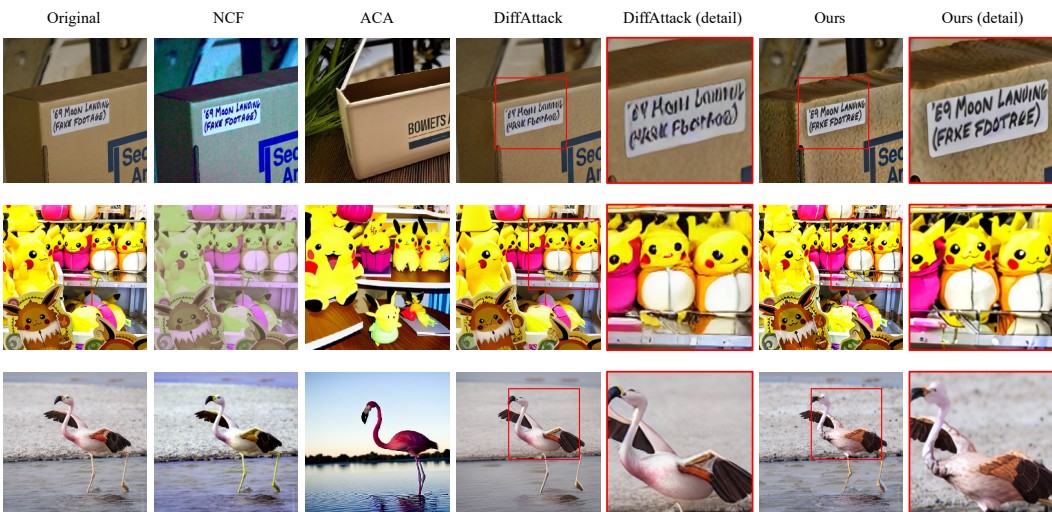

Figure 2: Visualization of unrestricted adversarial examples generated by state-of-the-art methods and our method. Since the image quality generated by our method is similar to that of DiffAttack (Chen et al., 2023), we provide zoomed-in images to compare the two methods' performance in preserving visual details. Additional results are included in the supplementary material.

(2018); Kurakin et al. (2018a); Nie et al. (2022); Geirhos et al. (2018) and test whether the baseline methods, as well as the approach we propose, remain effective when confronted with these defenses.

Table 3 presents the performance of various unrestricted adversarial attack methods when encountering different defense strategies. The adversarial examples are generated based on ResNet-50. For input preprocessing defense methods, the adversarial examples are processed by the defense mechanism and then re-evaluated on ResNet-50; for adversarially trained models, the adversarial examples are directly transferred to the target models for testing. As shown in the table, previous defense methods can somewhat reduce the effectiveness of unrestricted adversarial examples, but the latest methods are increasingly exhibiting stronger attack capabilities. Compared to input preprocessing methods, adversarially trained models demonstrate greater generalization and perform better against unknown adversarial examples. However, even on the best-performing model, Ensemble-IncRes-V2, our proposed method still achieved a 52.40% attack success rate, surpassing NCF, DiffAttack, and ACA by 24.5%, 19.3%, and 3.3%, respectively.

## 4.5 ABLATION STUDY

**Effect of Various Hyperparameters.** Since our proposed loss function consists of three components, we introduce two scaling factors, $\beta$ and $\gamma$, to adjust the importance of each loss function. It is evident that the final adversarial performance is influenced by the values of $\beta$ and $\gamma$. Therefore, we conducted extensive ablation experiments to explore the impact of these factors on both adversarial performance and image quality.

Figure 3 shows how the white-box attack success rate, average black-box attack success rate, PSNR, and LPIPS change when varying $\beta$ and $\gamma$ with ResNet-50 as the surrogate model. It can be observed that increasing $\beta$ and $\gamma$ generally reduces the attack success rate while improving image quality. However, this is not always the case. For instance, when $\beta = 10.0$, increasing $\gamma$ from 0.01 to 0.1 raises the average black-box attack success rate from 71.23% to 71.38%. Similarly, when $\beta = 100.0$, increasing $\gamma$ from 0.01 to 0.1 improves the white-box attack success rate from 94.2% to 94.6%. Likewise, when $\gamma = 0.1$, increasing $\beta$ from 30.0 to 100.0 results in a rise in the white-box attack success rate from 94.4% to 94.6%. These results highlight the complex interplay between the three loss functions.

**Effect of Stable Diffusion Versions.** We also tested the impact of different Stable Diffusion versions on the generated results. Specifically, we selected SD v1-4, v1-5, and v2-1 for evaluation.

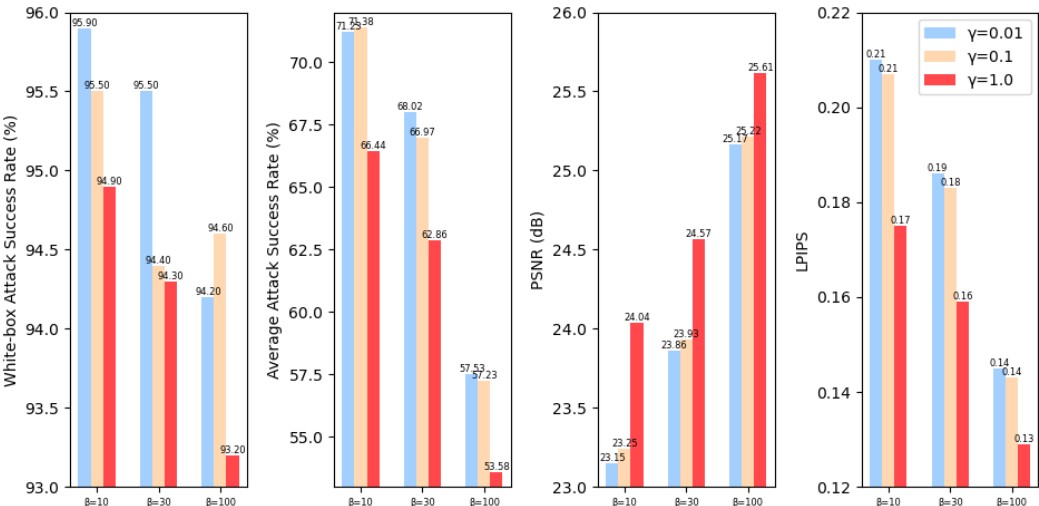

Figure 3: Impact of Hyperparameter Variations on the Performance of Generated Adversarial Examples.

Table 4: Comparison of the Impact of Different Stable Diffusion Versions on the Generated Results. Visualization results are included in the supplementary materials.

| SD version | White-box ASR↑ | Ave.ASR↑ | PSNR↑ | LPIPS↓ |
|---|---|---|---|---|
| v1-4 | 94.400 | 66.970 | 23.935 | 0.183 |
| v1-5 | 95.500 | 67.220 | 23.966 | 0.182 |
| v2-1 | 98.200 | 72.910 | 23.511 | 0.226 |

With all other hyperparameters held constant, Table 4 shows that the results generated by SD v1-5 outperform those of v1-4 across the board. In contrast, while SD v2-1 achieves a higher attack success rate, it produces images of lower quality. This indicates that different versions of Stable Diffusion may require distinct optimal parameter combinations for the best performance.

## 5 CONCLUSIONS

In this work, we propose an adversarial attacking method based on diffusion models. This method first maps the original image to a series of noise maps in the latent space through DDIM inversion and then performs adversarial optimization on the null-text embeddings using a null-text optimization method, thereby generating highly similar and transferable adversarial examples. To preserve the visual details of the generated results, we introduce perceptual feature reconstruction loss into our framework. Experiments demonstrate that our method can generate highly transferable adversarial examples, outperforming current state-of-the-art methods across multiple models, while also achieving best image quality. We hope this research draws attention to the security concerns of AI technologies.

**Limitations.** Due to the inherent limitations of diffusion models, a significant number of sampling steps are required during inference, causing our method to take around 60 seconds on an NVIDIA 6000 Ada. This makes it challenging to apply our proposed method in adversarial training at this stage.

**Social Impacts.** Our method can generate adversarial examples with strong black-box attack effectiveness and realism, which could be used for malicious attacks on deep learning models deployed on the internet or in specific industrial scenarios, thereby threatening high-security-sensitive applications.

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

# A APPENDIX

## A.1 MORE VISUALIZATION RESULTS

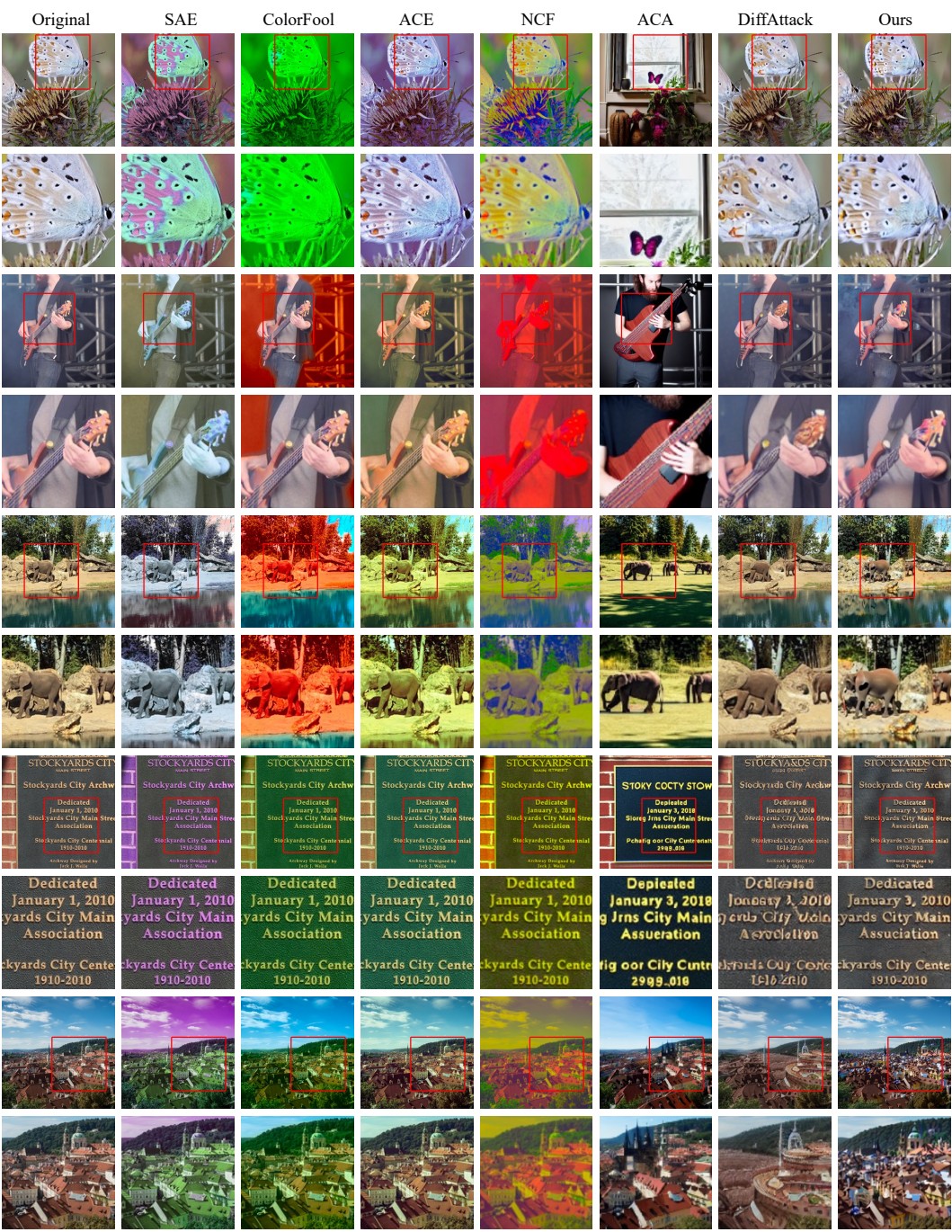

Figure 4: Visualization comparison of adversarial examples generated by various baseline methods. We also provide both full images and zoomed-in sections to illustrate the differences in detail between the adversarial examples produced by different methods.

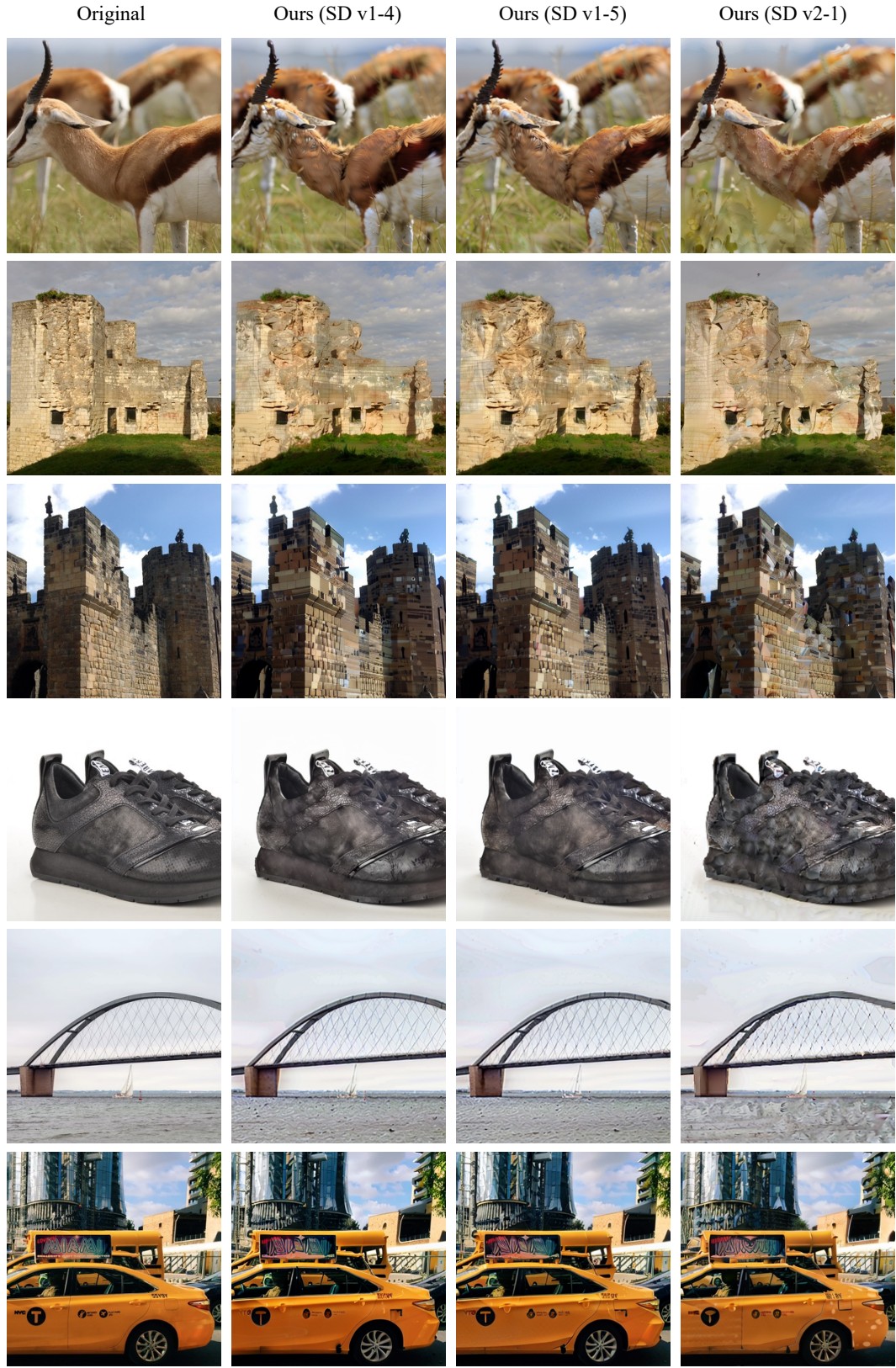

Figure 5: Differences in adversarial examples generated using different versions of Stable Diffusion. It can be observed that the results produced by v1-4 and v1-5 are similar, while the results generated by v2-1 are more distinctive, exhibiting noticeable artifacts.

## A.2 Attack performance in the physical world

To validate the feasibility of our attack method in the physical world, we printed both the original images and the generated adversarial examples, then photographed them again and input them into the classifier for recognition. All of the adversarial examples are generated based on ResNet-50. The classifier used for recognition is ResNet-50 as well. It can be observed that the adversarial examples we generated remain effective in the physical world. This clearly demonstrates the strong generalization and stability of our proposed attack method.

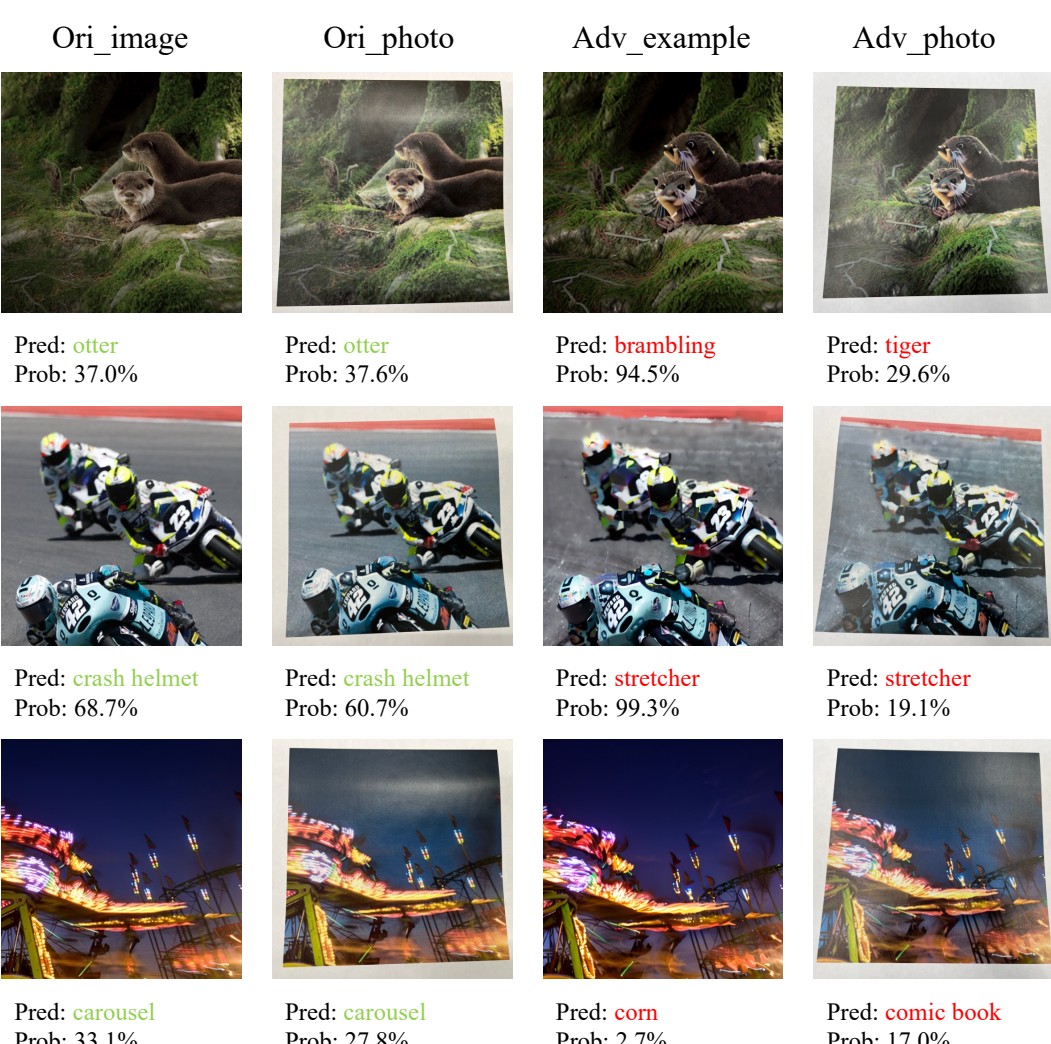

Figure 6: Predictions and probabilities of original image and adversarial examples in digital and physical world. Green and red represent the correct and incorrect prediction, respectively.

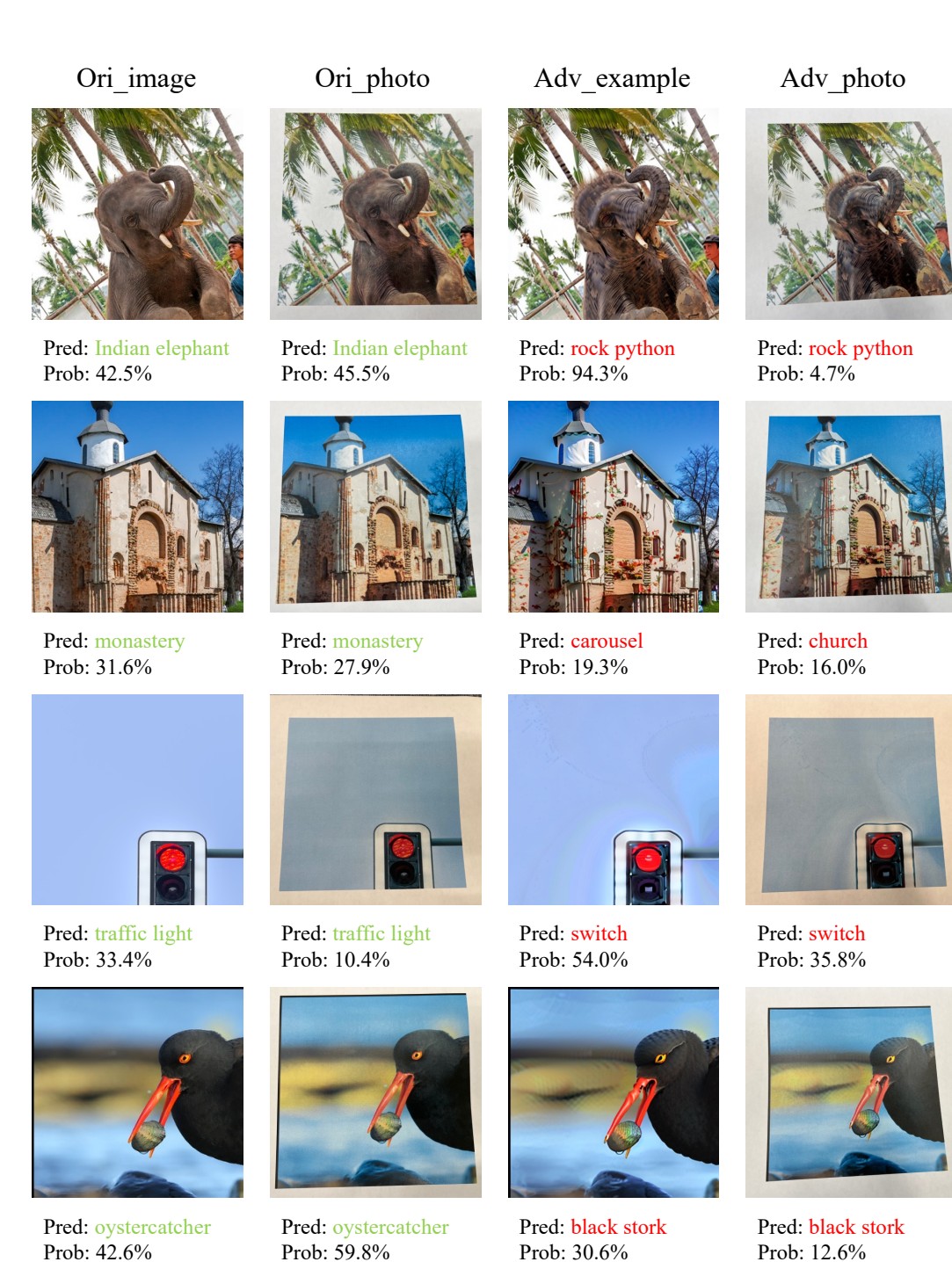

|  | Ori_image | Ori_photo | Adv_example | Adv_photo |
| --- | --- | --- | --- | --- |
| Pred: | Indian elephant | Indian elephant | rock python | rock python |
| Prob: | 42.5% | 45.5% | 94.3% | 4.7% |
| Pred: | monastery | monastery | carousel | church |
| Prob: | 31.6% | 27.9% | 19.3% | 16.0% |
| Pred: | traffic light | traffic light | switch | switch |
| Prob: | 33.4% | 10.4% | 54.0% | 35.8% |
| Pred: | oystercatcher | oystercatcher | black stork | black stork |
| Prob: | 42.6% | 59.8% | 30.6% | 12.6% |

Figure 7: More results in digital and physical world.

