# OpenReview forum: "More Harmful, Less noticeable: Learning Adversarial Null-Text Embeddings for Inconspicuous Attack"
_ICLR.cc/2025/Conference — ICLR 2025 Conference Withdrawn Submission_

### Official Review · Reviewer_txPF · 2024-10-25

**Soundness:** 3
**Presentation:** 3
**Contribution:** 3
**Rating:** 6
**Confidence:** 3

**Summary:**

This paper proposes an adversarial attack method based on the stable diffusion model. Unlike previous diffusion-based methods that directly optimize latent space features, the proposed method in this paper optimizes null-text embeddings to preserve the naturalness and fine-grained details of the generated adversarial samples. Experimental results show that the proposed method achieves improved transferability compared to existing approaches on both CNN- and ViT-based networks.

**Strengths:**

- This paper introduces a novel attack method that optimizes null-text embeddings instead of directly optimizing latent space features.
- Experimental results show that the generated adversarial samples exhibit better transferability than those produced by existing unrestricted adversarial example generation methods.
- Compared to previous attack methods, the proposed method achieves the best Attack Success Rates (ASR) across various defense mechanisms.

**Weaknesses:**

- The paper proposes to use feature reconstruction perceptual loss to enhance perceptual quality.  It would be helpful include experiment showing results with $\gamma=0$. This would provide a clear comparison point to understand the impact of the perceptual loss.

- Comparing computational costs (e.g.  average inference time per image and GPU memory usage) between the proposed method and existing approaches would be valuable in understanding its practical efficiency.

**Questions:**

Please see the weakness.

---

### Official Review · Reviewer_g6UF · 2024-10-28

**Soundness:** 2
**Presentation:** 3
**Contribution:** 2
**Rating:** 5
**Confidence:** 5

**Summary:**

This work further discusses the adversarial transferability of unrestricted adversarial examples. The authors believe that previous work modified the color or content of the image, which is easy to be noticed. Therefore, in this work, the authors introduce perturbations on text embedding based on stable diffusion and introduce perceptual loss in image reconstruction to minimize the change of content. Experiments show the effectiveness of the proposed method.

**Strengths:**

1. Exploring the adversarial transferability of unrestricted adversarial examples is helpful to better evaluate the robustness of existing deep neural networks.
2. Based on the diffusion model, this work constructs an adversarial attack in the image reconstruction process for the first time.
3. The generated unrestricted adversarial examples have relatively small modifications in the content.

**Weaknesses:**

1. I believe that the expression of Equation 2 is wrong. Unrestricted adversarial examples are not the same as $l_p$ norm. It is not enough to maximize the cross entropy. It is also necessary to ensure that the image is natural (defined in NCF and ACA). The most important constraint of unrestricted adversarial examples is missing here.

2.  The introduction of perceptual loss in adversarial attacks is not something novel. PPGD and LPA [1] have extensively discussed the application of perceptual loss in adversarial examples.

3. Further analysis is needed on the time cost relationship with existing methods.


[1] Perceptual adversarial robustness: Defense against unseen threat models, ICLR 2021

**Questions:**

1. Since unrestricted adversarial examples only need to ensure naturalness, why must we minimize the change of content? In principle, the generated adversarial examples can be natural to humans. We believe that this is a trade-off. If the degree of content-based attack is sacrificed, the adversarial transferability will definitely be reduced (also mentioned in lines 400-405). Therefore, this motivation needs to be further discussed and explained.

2. Table 2 mentions the image quality evaluation of FR and NR. Where is the image quality evaluation of NR? Unrestricted adversarial examples are mainly based on no-reference image quality assessment (NCF and ACA), which is very important.

---

### Official Review · Reviewer_DH5W · 2024-10-30

**Soundness:** 3
**Presentation:** 3
**Contribution:** 2
**Rating:** 5
**Confidence:** 4

**Summary:**

The paper introduces an adversarial attack method based on diffusion models, designed to produce visually high-quality and highly transferable adversarial examples. The approach applies adversarial optimization on null-text embeddings using a specialized null-text optimization technique to generate these adversarial examples. To enhance the fidelity of the generated images, the authors incorporate perceptual feature reconstruction loss. Experimental results demonstrate the proposed method's effectiveness in achieving superior transferability and image fidelity.

**Strengths:**

1, The experiments demonstrate the proposed method’s capability to achieve higher transferability and improved image fidelity.

2, The experiments cover a broader range of diverse victim model structures.

3, The paper is well-written and easy to understand.

**Weaknesses:**

1, The proposed method essentially applies a stable diffusion model to generate a specific type of image (adversarial images), which is not particularly novel.

2, The loss function used to improve image quality lacks an ablation study, making its effectiveness unclear.

3, The title of Table 2 mentions "FR" and "NR," but these terms are not used within the table.

**Questions:**

In Section 3.4, you mention that perceptual loss is used to improve image quality. Have you conducted an ablation study to demonstrate the specific benefit of this loss?

---

### Official Review · Reviewer_5jLH · 2024-11-03

**Soundness:** 2
**Presentation:** 2
**Contribution:** 2
**Rating:** 3
**Confidence:** 4

**Summary:**

This work aims to craft unrestricted adversarial examples with unnoticeable visual distortion and high adversarial transferability. To achieve these goals, this work proposes using stable diffusion and optimizing the null-text embeddings at certain denoising steps. Perceptual loss is also used to guarantee the visual quality of the perturbed samples when optimizing the adversarial examples. Experiments on the ImageNet subset reveal that their proposed method is effective at improving adversarial transferability, and shows good visual quality.

**Strengths:**

- This work proposes to perturb the null-text embeddings of diffusion models to craft the adversarial examples. This might be new to some extent (as perturbing the text modality has been studied to achieve adversarial image examples in existing works, see weakness).

- The authors conducted some experiments in the ImageNet subset, and experiments showed that, compared with some unrestricted attack baselines, this method shows good attack performance and good visual quality.

**Weaknesses:**

- Invalid claim of technical contribution. As shown in the Abstract, the authors claimed "we introduce perceptual loss into the adversarial attack process for the first time". Unfortunately, introducing perceptual loss has been studied for a long time ago, e.g. SSIM and LPIPS in [1,2]. Undoubtedly, utilizing this loss in an adversarial attack is nothing new at all. In the rebuttal, the authors may clarify whether you have missed related works (unintentionally), or there are some significant differences or challenges using the perceptual loss in diffusion models?

- While perturbing the null-text embedding of diffusion models seems new, crafting adversarial examples leveraging diffusion models is not new and has been studied by a lot of existing works (the authors also noted this aspect). Besides, attacking through perturbing the text modality has been studied too, e.g. [3], which should be discussed and used as a more recent baseline method. Given the fact that, the only technical contribution is perturbing the null-text embeddings of diffusion models, I feel the contribution may not be significant engough as a technical paper for ICLR. In the rebuttal, the authors are expected to explain how perturbing null-text embeddings differs from or improves upon these existing methods?

- Many baseline methods have been missing in the experimental comparison. For example, for unrestricted attacks, the authors are supposed to discuss and compare with state-of-the-art methods, e.g. [3,4]. BESIDES, regarding transferable comparison, the authors are strongly encouraged to compare with transferable attacks, e.g., MI-FGSM, TI-FGSM, S2I-FGSM etc, because they were specifically designed for such attacks.

- Some experimental results differ a lot from the ones reported in their original works. The authors are encouraged to double-check the experimental settings and discuss the reasons. For example, in Table 1, RN-50 row, the attack success rates of ACA differ a lot from the original work, ie, originally reported as 88.3%, but shown in this work as 72.10% for RN-50 white-box attack, which is a significant difference.  In the rebuttal, please provide a detailed explanation of the experimental setup, including any differences from the original papers, and discuss potential reasons for the discrepancies in results, such as differences in implementation, dataset preprocessing, or evaluation metrics.

- Time is another important factor in diffusion model-based attacks, therefore, the time comparison is also supposed to compare time taken to craft adversarial examples.

- The authors are also expected to provide some deeper analysis or insights regarding the feasibility of the null-text embedding attack, e.g., how and why they can reveal the adversarial vulnerability, and potential limitations or failure cases of the approach, etc.



[1] Jordan, Matt, Naren Manoj, Surbhi Goel, and Alexandros G. Dimakis. "Quantifying perceptual distortion of adversarial examples." arXiv preprint arXiv:1902.08265 (2019).

[2] Karli, Berat Tuna, Deniz Sen, and Alptekin Temizel. "Improving perceptual quality of adversarial images using perceptual distance minimization and normalized variance weighting." In The AAAI-22 Workshop on Adversarial Machine Learning and Beyond. 2021.

[3] Xu, Wenzhuo, Kai Chen, Ziyi Gao, Zhipeng Wei, Jingjing Chen, and Yu-Gang Jiang. "Highly transferable diffusion-based unrestricted adversarial attack on pre-trained vision-language models." In ACM Multimedia 2024. 2024.

[4] Dai, X., Liang, K., & Xiao, B. (2025). Advdiff: Generating unrestricted adversarial examples using diffusion models. arXiv 2023 (or ECCV 2024).

**Questions:**

See weakness.

**Details Of Ethics Concerns:**

NA.

---

### Note · Authors · 2024-11-13

I have read and agree with the venue's withdrawal policy on behalf of myself and my co-authors.